# The Effect of Endurance Training on Pulmonary $\dot{V}$O_2_ Kinetics in Solid Organs Transplanted Recipients

**DOI:** 10.3390/ijerph19159097

**Published:** 2022-07-26

**Authors:** Alessio del Torto, Carlo Capelli, Roberto Peressutti, Adriana Di Silvestre, Ugolino Livi, Chiara Nalli, Sandro Sponga, Giampaolo Amici, Umberto Baccarani, Stefano Lazzer

**Affiliations:** 1Department of Medicine, University of Udine, 33100 Udine, Italy; alessio.deltorto12@gmail.com (A.d.T.); ugo.livi@asufc.sanita.fvg.it (U.L.); chnalli@yahoo.com (C.N.); sandro.sponga@asuiud.sanita.fvg.it (S.S.); umberto.baccarani@uniud.it (U.B.); 2School of Sport Sciences, University of Udine, 33100 Udine, Italy; 3Department of Neurosciences, Biomedicine and Movement Sciences, University of Verona, 37131 Verona, Italy; carlo.capelli@univr.it; 4Regional Transplantation Centre, Friuli Venezia Giulia Region, 33100 Udine, Italy; roberto.peressutti@asuiud.sanita.fvg.it (R.P.); adriana.disilvestre@gmail.com (A.D.S.); 5Cardiac Surgery Unit, University Hospital of Udine, 33100 Udine, Italy; 6Nephrology and Dialysis Unit, San Daniele del Friuli Hospital, 33100 Udine, Italy; amici.gianpaolo@gmail.com; 7Liver-Kidney Transplant Unit, University Hospital of Udine, 33100 Udine, Italy

**Keywords:** pulmonary oxygen uptake kinetics, solid organ transplant, single-leg cycling, endurance training, slow component

## Abstract

Background: We investigated the effects of single (SL-ET) and double leg (DL-ET) high-intensity interval training on O_2_ deficit (O_2_Def) and mean response time (MRT) during square-wave moderate-intensity exercise (DL-MOD), and on the amplitude of $\dot{V}$O_2p_ slow component (SC_amp_), during heavy intensity exercise (DL-HVY), on 33 patients (heart transplant = 13, kidney transplanted = 11 and liver transplanted = 9). Methods: Patients performed DL incremental step exercise to exhaustion, two DL-MOD tests, and a DL-HVY trial before and after 24 sessions of SL-ET (*n* = 17) or DL-ET (*n* = 16). Results: After SL-ET, O_2_Def, MRT and SC_amp_ decreased by 16.4% ± 13.7 (*p* = 0.008), by 15.6% ± 13.7 (*p* = 0.004) and by 35% ± 31 (*p* = 0.002), respectively. After DL-ET, they dropped by 24.9% ± 16.2 (*p* < 0.0001), by 25.9% ± 13.6 (*p* < 0.0001) and by 38% ± 52 (*p* = 0.0003), respectively. The magnitude of improvement of O_2_Def, MRT, and SC_amp_ was not significantly different between SL-ET and DL-ET after training. Conclusions: We conclude that SL-ET is as effective as DL-ET if we aim to improve $\dot{V}$O_2p_ kinetics in transplanted patients and suggest that the slower, $\dot{V}$O_2p_ kinetics is mainly caused by the impairment of peripherals exchanges likely due to the immunosuppressive medications and disuse.

## 1. Introduction

Inadequate levels of fitness characterize heart, kidney, and liver transplant recipients (HTx, KTx, and LTx, respectively): their values of peak pulmonary O_2_ uptake ($\dot{V}$O_2p-peak_) are lower compared to their age-matched peers [1]. Although $\dot{V}$O_2p-peak_ is a strong predictor of cardiovascular mortality [2], we must consider that the most common daily activities are carried out at submaximal exercise intensities and require continuous transitions from one level of metabolic requirement to another. When the exercise is performed from rest to a constant submaximal load, the pulmonary O_2_ ($\dot{V}$O_2p_) kinetics reflects the skeletal muscle’s ability to promptly adjust the oxidative metabolism to sudden changes in metabolic requests. Indeed, when a constant load exercise is performed at moderate intensity (below the first ventilatory threshold (VT1)), $\dot{V}$O_2p_, after a first and sudden increase (phase I) due to the prompt uprise of pulmonary blood flow, gradually attains in about 3 min in young, healthy humans, the steady-state oxygen uptake ($\dot{V}$O_2ss_) following mono-exponential kinetics (phase II). For a given work rate, the faster the $\dot{V}$O_2p_ kinetics, the more rapidly $\;\dot{V}$O_2ss_ will be reached, and the lower intracellular perturbation (e.g., accumulation of H^+^ and depletion of PCr) will occur, a clue of a better muscular metabolic stability [3]. Hence, faster kinetics would result in a better performance capacity and a greater exercise tolerance [4].

For constant load exercise performed at heavy intensity (between VT1 and critical power/speed), $\dot{V}$O_2p_ requires a longer time to reach $\dot{V}$O_2ss_ because the so-called $\dot{V}$O_2p_ slow component ($\dot{V}$O_2pSC_) appears [5]. During this more prolonged phase characterized by a delayed adjustment of the oxidative metabolism, a substantial fraction of the total synthesized ATP derives from a more significant contribution of cytosolic substrate-level phosphorylation [6]. Also, in this case, the larger the transient accumulation of H^+^ and depletion of PCr will induce a more pronounced perturbation of the intramuscular milieu, heralding a quicker development of exhaustion. Therefore, reducing the $\dot{V}$O_2p_ slow component amplitude (SC_amp_) is another index of improved exercise tolerance [7].

In healthy individuals and normoxia, the speed of $\dot{V}$O_2p_ kinetics at the onset of moderate-intensity exercise is likely set by the muscular capacity of utilizing O_2_ rather than by muscle O_2_ delivery ($\dot{Q}$_m_O_2_). $\dot{V}$O_2p_ kinetics is decelerated when dysfunctions in the oxidative metabolism of skeletal muscle are present [8]. In addition, physio-pathological conditions that lead to reduced $\dot{Q}$_m_O_2_, may contribute to decelerating $\dot{V}$O_2p_ kinetics [9].

When exercising at heavy intensity, different muscular mechanisms are considered to be responsible for $\dot{V}$O_2pSC_ [5], such as the progressive recruitment of Type II muscle fibers [10,11], the metabolic processes occurring inside the already recruited fibers [12] and the impairment of O_2_ delivery during exercise [4]. 

Notably, transplanted recipients (Tx) are characterized by several muscle defects that may impair peripheral gas exchanges. Several concomitant causes induce these sequelae: f.i. immunosuppressive therapy and disuse/deconditioning may both negatively impact several of the factors that affect peripheral gas exchanges [1,13,14]. Indeed, in Tx $\dot{V}$O_2p_ kinetics is generally slower [15,16,17]. Notably, most of the studies focused their attention only on HTx [17,18,19] and few investigations addressed $\dot{V}$O_2p_ kinetics in LTx and KTx [15,16] and pointed out that the peripheral muscular factors were likely the main responsible for the slower speed of adjustment of $\dot{V}$O_2_. Interestingly, Tomczak and colleagues reported slower $\dot{V}$O_2p_ kinetics in both thoracic Tx (i.e., HTx) and abdominal Tx (i.e., KTx and LTx) [16]. This phenomenon was evident despite a decelerated heart rate (HR) response in HTx, suggesting lower kinetics of systemic oxygen delivery and a likely reduction of $\dot{Q}$_m_O_2_ [20]. Besides, $\dot{V}$O_2p_ kinetics were not accelerated in HTx after increasing the $\dot{Q}$_m_O_2_ using a priming exercise [18]. Altogether, these findings suggest that the peripheral impairment of gas exchanges, rather than the slower rate of adjustment of $\dot{Q}$_m_O_2_, might be the prevalent cause of the slower $\dot{V}$O_2p_ kinetics observed in HTx, LTx, and KTx.

Is it well known that endurance training (ET) elicits favorable adaptations at cardiovascular [21] and muscular [22] levels, speeding $\dot{V}$O_2p_ kinetics, and reducing SC_amp_ [7,23]. However, small muscle mass ET (i.e., SL or leg-kicking exercise) seems to be more effective than DL if one aims to improve the oxidative capacity of the skeletal muscle [24]. Moreover, leg-kicking training in elderly subjects resulted in the speeding of $\dot{V}$O_2p_ kinetics accompanied by a higher mitochondrial oxidative capacity, but without more significant muscle blood flow and muscle capillarization [25]. Furthermore, only one training study on HTx investigated the effect of exercise training on $\dot{V}$O_2p_ kinetics, disclosing that it was an effective tool for accelerating $\dot{V}$O_2p_ kinetics. Unluckily, this finding has not been confirmed yet in KTx and LTx [19]. Finally, no investigations have ever evaluated the effect of ET on SC_amp_ in HTx, LTx, and KTx.

Considering that: (i) disuse/deconditioning and immunosuppressant side-effects are common causes of the frequent muscular abnormalities found in HTx, KTx, and LTx; (ii) small muscle mass training leads to favorable adaptations to peripheral gas exchanges, we can hypothesize that peripheral factors affecting the muscle’s capacity of extracting and utilizing O_2_ might be the main responsible for the slower $\dot{V}$O_2p_ kinetics in transplant patients. Since small muscle mass training remarkably improves the factors affecting peripheral gas exchanges, we may hypothesize that ET involving a small muscle mass (i.e., SL) might be as effective as traditional whole-body ET (i.e., DL) if one aims to improve $\dot{V}$O_2p_ kinetics in this class of patients.

## 2. Materials and Methods

### 2.1. Participants and Anthropometric Characteristics

Thirty-eight sedentary Tx (n: 14, HTx; n: 13, KTx; n: 11, LTx) were included in this investigation. The procedures used in the current study were approved by the local Institutional Review Board (n: 8/IRB Department of Medical Area) and were carried out according to the Declaration of Helsinki. Subjects were recruited if at least one year had passed since the transplant operation. Before the study, participants were interviewed about their medical history, and afterward, they underwent a full medical check-up. The risks and possible side effects induced by the experimental procedures were explained to the subjects; their verbal and written consent was obtained. Their anthropometric characteristics were recorded; briefly, a mechanical scale (Seca 709, Hamburg, Germany) was used to assess body mass, to the closest 0.1 kg, with the patients wearing only light underwear and stockings. A stadiometer applied to the wall was used to measure stature to the closest 0.5 cm. The ratio calculated by dividing the body mass (body mass, kg) and the squared stature (m) was used to obtain the body mass index. If patients were diagnosed with cancer, cardiorespiratory and orthopedic diseases that could have prevented participation in the endurance training protocol were excluded from the study; pregnant participants were also excluded. 

A total of five patients withdrew from the study before the start of the endurance training phase; therefore, 33 subjects (male = 28; female = 5) were enrolled and completed the investigation (HTx = 13, KTx = 11, and LTx = 9). 

### 2.2. Study Design

Following the first visit, participants were randomly assigned to SL (SL-ET_GRP_; *n* = 17) and DL (DL-ET_GRP_; *n* = 16) endurance training groups. During the two weeks preceding the beginning of the ET program (PRE), the subjects were tested to determine their cardio-respiratory and anthropometric parameters; moreover, subjects were invited to the laboratory to familiarize themselves with the experimental procedures before the first testing session. One to two days after the end of the ET program (POST), the participants performed the same battery of tests that they carried out at PRE to determine the modifications in the investigated outcomes.

The protocol included three experimental sessions performed on different days divided by 2 days. Participants were advised to refrain from vigorous exercise the day preceding each testing session, avoid beverages containing caffeine (5 h), and food consumption at least 3 h before every laboratory visit. On the first day, patients underwent medical screening and anthropometric assessment; afterward, they performed a DL-INC test. During the following visit, participants completed a DL-MOD trial, and on the third visit, they carried out two additional tests. Firstly, DL-MOD repetition and, secondly, a DL-HVY test. The two exercise bouts were separated 60- of rest. 

### 2.3. Double Leg Incremental Step Test

After 3 min of rest in a sitting position on the ergometer, the participants started cycling for 6 min at 25 W or 40 W as a warm-up. Afterward, a DL-INC began, and the mechanical resistance was increased by 15 W each minute until the tolerance limit was reached. The pedaling frequency was displayed on the ergometer screen, and participants had to maintain a constant cadence at their preferred rate between 60 and 75 rpm. The test was completed when the patients reached exhaustion or when they could not keep the pedaling frequency above 60 rpm despite verbal incitation.

Immediately after the conclusion of the test, the mechanical power was lowered to 25 W, kept constant for five minutes to cool down, and RPE_leg fatigue_ and RPE_dyspnea_ were recorded during this phase [26]. The highest mean values obtained from subsequent 30-s epochs represented the $\dot{V}$O_2p-peak_, $\dot{V}$CO_2p-peak_, $\dot{V}$_Epeak,_ and HR_peak_. The peak mechanical power was considered as the one associated with the last completed workload. However, if the exhaustion was reached before the completion of a 1-min stage, the associated workload was computed as previously explained [27].

Two experienced researchers, familiar with the procedure, determined the $\dot{V}$O_2_-VT1 adopting the V-slope method; however, additional criteria were also used, namely $\dot{V}$_E_/$\dot{V}$O_2_ and $\dot{V}$_E_/$\dot{V}$CO_2_ and P_ET_O_2_ and P_ET_CO_2_ [28]. If the two assessors disagreed, a third trained researcher performed the analysis adopting the three criteria until agreement was reached and $\dot{V}$O_2_-VT1 determined. 

Moreover, to calculate the workload associated with $\dot{V}$O_2_-VT1, the $\dot{V}$O_2p_ vs. power relationship was shifted to the left by 30 s for every subject.

$\dot{V}$O_2p_, $\dot{V}$CO_2p_, and $\dot{V}$_E_ were determined breath-by-breath by a metabolic cart (CPET, Cosmed, Italy), and the ECG signal was used to measure the HR. O_2_ and CO_2_ analysers were calibrated, by utilizing a gas mixture of known composition (16.00% O_2_, 4.00% CO_2_, nitrogen as the balance), before the beginning of each experimental session. Furthermore, the turbine flowmeter was calibrated by using a 3 L syringe with three different flow rates.

### 2.4. Constant Load Exercises

In the second visit, the volunteers performed a DL-MOD exercise test at a work rate corresponding to 80% of the power output at VT1. After having positioned the ECG electrodes, volunteers rested for 3 min sitting on the cycle-ergometer. Then they started cycling at a constant work rate for 10 min and were asked to keep a constant pedaling cadence at their preferred cadence (60–75 RPM), which was digitally displayed and recorded for the subsequent experimental sessions. Once the bouts were concluded, the RPE_leg fatigue_ and RPE_dyspnea_ were registered. Gas exchanges, $\dot{V}$_E_ and HR were continuously measured throughout the test. 

In the third experimental session, the DL-MOD was repeated. After this exercise bout, volunteers rested sitting on the bike for ~15 to 30 min. Afterward, they performed a DL-HVY test at a work rate corresponding to 40% of the difference between $\dot{V}$O_2_-VT1 and $\dot{V}$O_2p-peak_. Participants were asked to pedal until exhaustion; otherwise, the exercise was interrupted after 14 min. Also, in this case, RPE_leg fatigue_ and RPE_dyspnea_ were registered at the end of the test. Blood lactate concentration ([La]^b^) was assessed by an enzymatic method (Biosen C-line; EKF) at rest, at the 1st min after the DL-MOD cessation, and at the 1st, 3rd, and 5th min after the end of DL-HVY. The highest value registered during the recovery phase was retained as peak [La]_b_.

### 2.5. Endurance Training

The two training groups completed 8 weeks of ET divided into 3 sessions per week, which research assistants always supervised. High-intensity interval training (HIIT) was adopted as a training modality because it was previously shown to be beneficial and well tolerated by this class of patients [29]. To increase training compliance, two different HIIT schemes were adopted [22,29]. 12 training sessions involved 4 min at high intensity alternated by 3 min of low intensity, and each interval was performed 4 times. The other 12 training sessions involved 2 min at high intensity alternated by 2 min of low intensity. With this scheme, each interval was performed 6 times. Subjects were asked to complete both the two HIIT protocols alternately. The HIIT sessions started with 5 min of light cycling as a warm-up and concluded with 5 min of light pedaling to cool down. DL pedaling was carried out for the whole session by the DL-ET group. On the other hand, the SL-ET group performed only single-legged cycling; during each ET session, they completed the first half of the high and low-intensity intervals with one lower limb and the remaining half cycling with the other limb. 

This structure allowed the SL-ET and the DL-ET to train for an identical amount of time. Moreover, both groups were instructed to maintain a constant pedaling frequency, within 60 to 75 RPM, during the HIIT bouts. Therefore, the amount of mechanical work carried out by the musculature of the lower limbs was very comparable for the DL-ET and SL-ET.

10 kg were applied to the opposite crank for the SL training, and a wooden structure was positioned close to the ergometer as a support and shield for the resting leg [24]. The RPE scale was used to adjust the intensity of the HIIT bouts; the mechanical resistance for the high-intensity stages was set to reach a value equal to or greater than 15 of RPE_dyspnoea_ during the DL-ET and equal to or higher than 5 of RPE_leg fatigue_ during the SL-ET [30].

The low-intensity stages were set to obtain a score equivalent to or inferior to 12 on RPE_dyspnoea_ for the DL-ET and equal to or lower than 2 on RPE_leg fatigue_ for the SL-ET.

HR was recorded for the whole training session as the mean values prevailing in the last 15 sec of each high and low-intensity stage; by the same token, RPE_dyspnoea_ and RPE_leg fatigue_ were measured once each high and low-intensity bout were completed. The work rates for the HIIT were modified after every week of ET using the RPE [30,31] and the research assistants supervised all the patients during each HIIT session to ascertain that at least 90% of the workouts were executed correctly.

### 2.6. Data Treatment

$\dot{V}$O_2p-ss_, $\dot{V}$CO_2p-ss_, $\dot{V}$_Ess,_ and HR_ss_ were computed by averaging breath-by-breath values assessed in the last 2 min of the two DL-MOD and the last minute of exercise during the DL-HVY. [La]^b^ at the end of DL-MOD was calculated as the average of the two values obtained at the end of the DL-MOD tests.

We did not perform the fitting of $\dot{V}$O_2p_ vs. time to describe kinetics analysis. Instead, we calculated the oxygen deficit (O_2_Def) and the mean response time (MRT) as previously reported [32]. This approach provides MRT values that are identical to those obtained by fitting the same response with a simple exponential function without time delay [33]. Briefly, B-by-B $\dot{V}$O_2p_ values of each DL-MOD repetition were interpolated to 1-s intervals [34], time aligned with the onset of the exercise test, and treated by subtracting the $\dot{V}$O_2p_ at rest. The two repetitions’ data were then combined to obtain a single data file for each subject and condition. O_2_Def was calculated as the difference between the O_2_ that would have been consumed if $\dot{V}$O_2p-ss_ had been attained immediately at the beginning of the exercise and the volume of O_2_ taken up during exercise. The first quantity was calculated by multiplying $\dot{V}$O_2p-ss_ in ml O_2_ s^−1^ by the exercise duration (600 s). The O_2_ volume consumed during exercise was calculated by summing progressively the $\dot{V}$O_2p_ values expressed in ml O_2_·s^−1^ from the trial’s onset to 600 s. MRT was computed as the ratio between O_2_Def and the corresponding $\dot{V}$O_2p-ss_. Gross pedaling efficiency was calculated according to Péronnet and Massicotte [35], and the oxygen cost of cycling (O_2_cost) was computed by dividing the difference between $\dot{V}$O_2p-ss_ and resting $\dot{V}$O_2_ for the corresponding W. 

To detect $\dot{V}$O_2pSC_ during DL-HVY, we averaged and linearly fitted the $\dot{V}$O_2p_ values calculated every 30 s from the third to the sixth of the exercise. A positive slope significantly different from 0 would indicate the development of the $\dot{V}$O_2pSC_ [36]. Moreover, the SC_amp_ during the DL-HVY was determined as the difference between the $\dot{V}$O_2p_ at the 3rd min and that at the last minute of the exercise [37]. $\dot{V}$_E_ values were averaged every 30 s and used to estimate the work of breathing (WB) and the oxygen cost of respiratory muscles ($\dot{V}$O_2RM_) as [6]:(1)$$\mathrm{WB}=-0.430+0.050\cdot ({\dot{V}}_{E})+0.00161\cdot ({\dot{V}}_{E}{)}^{2}$$
(2)$$\dot{V}O_{\text{2-RM}}=(34.9+7.45\cdot \mathrm{WB})$$

HR values coinciding with every single DL-MOD repetition were time aligned with the exercise onset and superimposed, then the HR values were averaged each 5-s epoch. Afterward, the data were fitted by the function:(3)$$y\left(t\right)=y_{BAS}+A_{f}[1-e^{(t-TD_{f}{)/\tau }_{f})\;}]$$
where y(*t*) represents HR as a function of time *t*; $y_{BAS}$ is the baseline value of HR; $A_{f}$ is the amplitude of the fundamental component of the response between baseline HR and HR at steady-state; $TD_{f}$ is the time delay, and $\tau_{f}$ is the time constant of the function for the fundamental component (HR_tau_).

### 2.7. Statistics

Prism, version 8.0 (GraphPad Software, La Jolla, CA, USA), was used for the data analysis. Tables and text report the data as means ± standard deviation (SD) unless stated otherwise; moreover, the mean difference is expressed as PRE minus POST values. The Shapiro-Wilk test was used to determine whether the data were normally distributed. Cardio-respiratory parameters, HR, and $\dot{V}$O_2p_ kinetics were studied before and after the ET intervention; a two-way, within-subject ANOVA was used for the statistical analysis, in which time was identified as PRE- and POST- training and group as SL-ET_GRP_ and DL-ET_GRP_. Post hoc Bonferroni’s multiple comparisons test was carried out when the analysis had a significant main effect or interaction effect. Paired Student’s *t*-test was used to compare the 30 s averaged $\dot{V}$O_2p-ss_ within the two training groups during DL-MOD. The differences between SL-ET_GRP_ and DL-ET_GRP_ in their anthropometrical variables before training and the number of ET sessions performed by each group were analyzed with unpaired Student’s *t*-tests. Linear regressions were calculated by the least-squared residuals method [38], and the difference between slopes was evaluated as indicated by Zar [39]. Partial eta squared (ηp^2^) is reported according to del Vecchio and colleagues [40] for the main effect of time, except as otherwise indicated; moreover, the tables contain the mean difference for the PRE and POST parameters and their 95% confidence interval. Alpha level was set to ≤0.05, and values between >0.05 and ≤0.10 were considered to indicate trends.

The HR model parameters were estimated using an iterative, weighted nonlinear least-squares procedure [41] implemented by the commercial software for data analysis Prism, version 8.0 (GraphPad Software, La Jolla, CA, USA). We entered initial guesses of the model’s parameters after visually inspecting the data. 0.08 was the power value obtained from the calculation (G*Power) for the post-hoc analysis, which was computed including the 33 subjects, and 0.25 as size effects, for the primary outcomes (O_2_Def and MRT).

## 3. Results

### 3.1. Patient Characteristics and the Exercise Training Regimen

Three participants in the SL-ET_GRP_ and one in the DL-ET_GRP_ completed only one DL-MOD. Therefore, we decided to exclude their data from the analysis of $\dot{V}$O_2p_ and HR kinetics. The main anthropometric characteristics and peak cardio-respiratory parameters at PRE are reported in Table 1. The pharmacological therapies for the SL-ET_GRP_ and DL-ET_GRP_ are reported in Table 2. 

### 3.2. Double Leg Moderate Constant Load Test

Table 3 and Table 4 report the data related to $\dot{V}$O_2p_ and HR kinetics and the data measured at steady-state, respectively. $\dot{V}$O_2p_ and HR kinetics parameters for HTx and non-cardiac Tx are shown in Appendix A. The data of Tx treated with β-blocker medications and of the patients not treated with β-blockers are shown in Appendix A.

At PRE HR_ss_ (*p* = 0.016) was slightly, but significantly, higher in DL-ET_GRP_ (Table 4). HR at baseline and the response amplitude of HR kinetics tended to be lower in SL-ET (*p* = 0.064 and *p* = 0.098, respectively) (*p* = 0.061) (Table 3). 

The $\dot{V}$O_2p-ss_ assessed during DL-MOD corresponded to 92 ± 6% and 92 ± 6% of $\dot{V}$O_2_-VT1 for SL-ET_GRP_ and DL-ET_GRP_, respectively. Moreover, the 30-s averaged $\dot{V}$O_2p_ calculated during the third min (from 180th s to the 210th s) of the exercise was not significantly different from the one calculated during the sixth min in SL-ET_GRP_ and DL-ET_GRP_ (*p* = 0.306 and *p* = 0.517, respectively). This finding confirmed that $\dot{V}$O_2p_ attained the steady-state. $\dot{V}$_Ess_ tended to decrease by 4.7% (6.5) in SL-ET_GRP_ (*p* = 0.065) and was significantly lower by 5.4% ± 7.6 in DL-ET_GRP_ (*p* = 0.017). Moreover, HR_ss_ was also significantly reduced by 5% ± 7.5 and 8.5% ± 5.2 in SL-ET_GRP_ (*p* = 0.011) and DL-ET_GRP_ (*p* < 0.0001), respectively (Table 4). The change in $\dot{V}$_Ess_ was not statistically different between the two groups (*p* = 0.154). HR_ss_, tended to assume a lower value in DL-ET_GRP_ compared to SL-ET_GRP_ (interaction effect (G × T), *p* = 0.094). Finally, [La]^b^ was significantly reduced by 25% ± 18 in SL-ET_GRP_ (*p* = 0.0003) and by 37% ± 17 in DL-ET_GRP_ (*p* = < 0.0001); the changes tended to be different between the two groups (G × T, *p* = 0.057) (Table 4).

At POST, O_2_Def significantly decreased by 16.4% ± 13.7 (*p* = 0.008) and 24.9% ± 16.2 (*p* < 0.0001) in SL-ET_GRP_ and DL-ET_GRP_, respectively. Likewise, MRT dropped by 15.6% ± 13.7 (*p* = 0.004) (*p* = 0.004) in SL-ET_GRP_ and by 25.9% ± 13.6 (*p* < 0.0001) in the DL-ET_GRP_ (Table 3). The changes in O_2_Def and MRT were not significantly different between the two groups (*p* = 0.277 and *p* = 0.083, respectively). HR_Tau_ decreased by 32% ± 22, in SL-ET_GRP_ (*p* = 0.001), and by 23% ± 41 DL-ET_GRP_ (*p* = 0.006); no differences were found between the two groups (Table 3). r^2^ of the fitting ranged from 0.94 to 0.99 and from 0.95 to 0.99 for both groups at PRE and POST, respectively. 

### 3.3. Double Leg Heavy Constant Load Test

Table 5 shows the refers to DL-HVY. The slopes of the linear relationship between the $\dot{V}$O_2p_ vs. time were significantly different from zero in SL-ET_GRP_ (*p* < 0.0001) and in DL-ET_GRP_ (*p* < 0.0001) before training (PRE), indicating the presence of $\dot{V}$O_2pSC_ (Figure 1). Only one subject was not able to reach the 6th min of DL-HVY. Therefore, he was excluded from the slope analysis. In the SL-ET_GRP_, 15 volunteers exercised up to the 8th, 14 up to the 9th, and 12 up to the 14th minute of exercise; in the DL-ET_GRP_, 15 subjects reached the 8th, 14, and 9th, and 12 the 14th min of exercise.

All subjects in DL-ET_GRP_ could reach the 14th min of DL-HVY at POST, whereas in SL-ET_GRP_ only one volunteer reached exhaustion at the 7th min of the trial. In SL-ET_GRP_, SC_amp_ decreased by 35% ± 31 (*p* = 0.002); likewise, it dropped by 38% ± 52 (*p* = 0.0003) in DL-ET_GRP_. The changes were not significantly different between the training groups (*p* = 0.654) (Table 5). The angular coefficients of the linear relationship between $\dot{V}$O_2p_ and time were statistically different from zero in SL-ET_GRP_ and DL-ET_GRP_ also at POST (*p* = 0.0004 and *p* = 0.0002, respectively) (Figure 1); they were also significantly lower than the ones prevailing at PRE in both SL-ET_GRP_ and DL-ET_GRP_ (*p* < 0.0001 and *p* = 0.0012, respectively) (Figure 1). Moreover, the two slopes were not significantly different at POST (*p* = 0.676).

$\dot{V}$O_2-RM_ significantly decreased by 13% ± 19 and by 23% ± 15 in SL-ET_GRP_ and DL-ET_GRP_, respectively (*p* = 0.023 and *p* = 0.0003, respectively) (Table 5). To assess if $\dot{V}$O_2-RM_ substantially contributed to $\dot{V}$O_2pSC_, “gross” $\dot{V}$O_2p_ was correct by subtracting the estimated $\dot{V}$O_2-RM,_ and we calculated again the slope of the regression lines from the 3rd to the 6th min. The angular coefficients of the corrected $\dot{V}$O_2p_ vs. time were not significantly different from the ones obtained from the “gross” $\dot{V}$O_2p_ vs. time in SL-ET_GRP_ (*p* = 0.299) and DL-ET_GRP_ (*p* = 0.496). Besides, the slope of the corrected $\dot{V}$O_2p_ vs. time was not significantly different between DL-ET_GRP_ and SL-ET_GRP_ (*p* = 0.370) (Figure 2).

$\dot{V}$O_2p_ at the last min of DL-HVY decreased by 4.1% ± 5 (*p* = 0.012) and by 6.3% ± 7.5 (*p* = 0.001) in SL-ET_GRP_ and DL-ET_GRP_, respectively; there was no difference between the changes in the two groups (*p* = 0.469; Table 5). At POST, $\dot{V}$_E_ was 10.6% ± 10.4 (*p* = 0.001) and 17.1% ± 9.3 lower (*p* < 0.0001) in SL-ET_GRP_ and DL-ET_GRP_ than before training, respectively (Table 5). O_2_ cost decreased, although not significantly so, by 4.1% ± 5.8 (*p* = 0.181) in SL-ET_GRP_; it significantly dropped by 6.7% ± 8.3± (*p* = 0.002) in DL-ET (Table 5). Neither $\dot{V}$_E_ nor O_2_cost were significantly different between the two groups (*p* = 0.158 and *p* = 0.160, respectively) (Table 5).

HR at the end of DL-HVY significantly decreased by 6.2% ± 5.3 (*p* = 0.002) and 9.1% ± 6.7 (*p* = 0.002) in SL-ET_GRP_ and DL-ET_GRP_, respectively; a tendency towards a difference was found between the two groups (*p* = 0.098, Table 5). [La]^b^_peak_ statistically dropped in SL-ET_GRP_ and DL-ET_GRP_ by 26% ± 35 (*p* = 0.026) and 63% ± 70 (*p* < 0.0001), respectively; moreover, the difference between the two groups tended to be significant (*p* = 0.088, Table 5).

## 4. Discussion

In the current study, we compared for the first time the effect of 8 weeks of single–leg and double-leg HIIT on $\dot{V}$O_2p_ and HR kinetics in different groups of transplanted patients.

The main results showed that: (i) SL-ET was effective in improving $\dot{V}$O_2p_ and HR kinetics during moderate intensity exercise and in reducing the amplitude of $\dot{V}$O_2pSC_ during heavy intensity exercise; (ii) no difference between SL-ET_GRP_ and DL-ET_GRP_ were found as for the improved $\dot{V}$O_2p_ and HR kinetics during moderate intensity exercise and as for the attenuated SC_amp_ and reduction of the slopes of the linear increase of $\dot{V}$O_2p_ during heavy intensity exercise.

Given that our group of patients was heterogeneous, as it included heart, liver, and kidney transplant recipients, it can be argued that the different types of transplants received may elicit different physiological responses to exercise and training. Indeed, HTx recipients, because of cardiac denervation, are characterized by the so-called chronotropic incompetence [1]. This condition is often responsible for a higher HR at rest, a lower HR_peak_, and a sluggish HR adjustment during constant-load exercise that might prevent the attainment of adequate values of $\dot{Q}$_a_O_2_ during submaximal exercise and hence reduce the magnitude of $\dot{Q}$_m_O_2_ throughout the transient phase at the onset of the exercise. In turn, the sluggish response of $\dot{Q}$_m_O_2_ may contribute to decelerating $\dot{V}$O_2p_ kinetics at the beginning of DL-MOD. Indeed, HTx present a slow $\dot{V}$O_2p_ kinetics [18,19,21].

The slower HR kinetics in our patients may suggest a parallel slower rate of adjustment of $\dot{Q}$_m_O_2_ [42], contributing to the deceleration of $\dot{V}$O_2p_ kinetics in HTx compared to non-cardiac Tx. However, despite the decelerated HR kinetics, a previous study reported that the $\dot{V}$O_2p_ rate of adjustment was not different between HTx and non-cardiac Tx [16]. It is worth noting that the increase of $\dot{Q}$_m_O_2_ obtained by priming exercise turned out to be ineffective in speeding $\dot{V}$O_2p_ kinetics in HTx during the subsequent constant load exercise transition [18]. Our data confirmed this matter of fact: although HR kinetics in HTx was significantly slower (*p* = 0.001) than in the other transplant recipients, O_2_Def and MRT were not different between HTx and non-cardiac transplanted patients (*p* = 0.406 and *p* = 0.531, respectively) (see Appendix A). Besides, when Tx treated with β-blockers were compared with not treated patients, no differences were detected for HR_Tau_, O_2_Def, and MRT between the two groups (*p* = 0.995, *p* = 0.672, and *p* = 0.556, respectively) (see Appendix A). These findings indicate that neither the cardiac denervation nor the β-blockade treatment has negatively influenced, *per se*, the rate of adjustment of the oxidative metabolism at the exercise onset in our Tx. Considering that the cardiac denervation affects only HTx, we can hypothesize that additional mechanisms elicited the acceleration of $\dot{V}$O_2p_ kinetics and the attenuation of SC_amp_ attenuation found in HTx.

In the present study, however, we cannot fully disentangle the possible physiological mechanisms underpinning the improvement of $\dot{V}$O_2p_ kinetics at moderate and heavy exercise intensities in HTx and solid organ transplanted patients. The absence of skeletal muscle biopsies and of the determination of $\dot{Q}$_m_O_2_ prevented us from clarifying if different central or peripheral adaptations occurred in HTx and non-cardiac Tx. Yet, it must be highlighted that the absolute changes of the SC_amp_, O_2_Def, and MRT were of the same magnitude in the two classes of transplant recipients. Namely, (i) SC_amp_ decreased by 22% (38) in cardiac and by 45% (43) in non-cardiac Tx, with no difference between them (*p* = 0.13); (ii) O_2_Def decreased by 23.6% (15.3) in cardiac and by 16.2% (15.2) in non-cardiac Tx, with no difference between the groups (*p* = 0.212) and; (iii) MRT improved by 16.4% (14.1) in cardiac and by 16.2% (15.2) in non-cardiac Tx, with no difference between the types of transplants (*p* = 0.188). Therefore, this data allowed us to compare DL-ET_GRP_ and SL-ET_GRP_ since heterogeneous groups of Tx formed them.

The results suggest that ET effectively induced beneficial adaptations of $\dot{Q}$_m_O_2_ or m$\dot{V}$O_2,_ and they also indicate that, regardless of the type of transplant, the $\dot{V}$O_2p_ kinetics was accelerated and SC_amp_ smaller. The following paragraphs discuss the possible mechanisms behind the observed improvements.

### 4.1. $\dot{V}$O_2p_ Kinetics Parameters and Moderate-Intensity Exercise

Previous studies showed ET speeded $\dot{V}$O_2p_ kinetics during moderate exercise in older and young healthy subjects [14,27]. We confirmed and extended these findings to HTx, LTx, and KTx. Indeed, MRT and O_2_Def were significantly reduced in SL-ET_GRP_ and DL-ET_GRP_.

Our results agree with those reported by Tomczak et al. (2013), who showed a faster phase II $\dot{V}$O_2p_ kinetics in a group of HTx following 12 weeks of combined endurance and strength training [19]. Besides, the faster $\dot{V}$O_2p_ kinetics found in the SL-ET_GRP_ agrees with the results presented by Bell and colleagues [25]. They reported accelerated $\dot{V}$O_2p_ kinetics in the trained leg of elderly subjects after knee extension training. As already outlined, faster $\dot{V}$O_2p_ kinetics and a reduced O_2_Def also lead to a lower perturbation of the intracellular milieu, thus contributing to better exercise tolerance [4]. Therefore, we may surmise that the ameliorations of MRT and O_2_Def induced by SL-ET and DL-ET led to improved exercise tolerance in the evaluated volunteers. Also, the lower [La]^b^, HR_ss,_ and $\dot{V}$_Ess_ seem to indicate reduced cardio-respiratory and metabolic stress after training.

The initial differences in HR_ss_, at baseline and the HR amplitude response may be caused by the different number of patients under β-blockade medications in the two groups. Nevertheless, the speeding of $\dot{V}$O_2p_ kinetics was accompanied by a faster HR_Tau_ in all groups of transplanted volunteers suggesting a more rapid $\dot{Q}$_m_O_2_ response at the onset of DL-MOD [42]. Yet, the improved m$\dot{V}$O_2_ cannot be identified as the exclusive mechanism behind the faster $\dot{V}$O_2p_ kinetics, as the improvement of the peripheral gas exchanges after training cannot be ruled out as an additional beneficial mechanism.

Single leg training has been described to induce specific muscle adaptations that result in improved peripheral gas exchanges [24]. The data collected at PRE and POST during DL-MOD (in parallel with the ones discussed in the next paragraph concerning heavy intensity exercise) seem to confirm that in these patients, the muscular capacity of extracting and utilizing O_2_ is particularly jeopardized. A training modality that modifies the phenotypic expression of the recruited muscles, but hardly influences the cardiovascular capability of transporting O_2_ to the periphery, was capable of inducing beneficial effects on $\dot{V}$O_2p_ kinetics. Yet, as neither O_2_cost nor gross pedaling efficiency significantly changed after ET, we cannot exclude that the oxidative energy-yielding pathway of the muscles was not fully adapted after training [43,44].

### 4.2. Exercise Responses to Heavy Intensity Exercise

To our knowledge, this is the first study that evaluated the effect of ET, and more specifically SL-ET vs. DL-ET, on SC_amp_ in transplanted patients. The reduction of SC_amp_ is a well-described adaptation to ET, and the reported drop of SC_amp_ and the lower slope of the linear regression of $\dot{V}$O_2p_ vs. time at POST, when compared to the one assessed at PRE, agree with the findings of other investigations [45]. Moreover, the reduction of the SC_amp_ was not different between SL-ET_GRP_ and DL-ET_GRP,_ and the slopes of $\dot{V}$O_2pSC_ were not different between the two groups at POST. These findings indicate that the SL-ET was as effective as the DL-ET in causing the favorable adaptations responsible for decreasing $\dot{V}$O_2pSC_.

Recent findings have suggested that the progressive recruitment of the less economic Type II fibers maybe not be strictly necessary to induce $\dot{V}$O_2pSC_. In contrast, it may derive from mechanisms inherent to the recruited fibers [12] responsible for increasing the O_2_ cost of oxidative synthesis of ATP. A recent study [46] showed that endurance training in rats induced a temperature-dependent enhancement of mitochondrial oxidative phosphorylation and a significant drop in mitochondrial uncoupling. Therefore, the decrease of O_2_ cost for oxidative ATP production in each recruited muscle fiber may have substantially potentiated the effect of endurance training on $\dot{V}$O_2pSC_. Accordingly, the decreased [La]^b^_peak,_ and the improved exercise economy supports the view of improved oxidative metabolism after endurance training [47]. However, we cannot disentangle whether this intrinsic muscular adaptation was more responsible than the change of the pattern of motor-unit recruitment in eliciting $\dot{V}$O_2pSC_.

Even though muscular mechanisms account for more than 80% of SC_amp_ [5], extrinsic factors such as the O_2_cost of respiratory muscles [48], cardiac work, and auxiliary muscles’ contractions may explain the remaining fraction [49]. Indeed, Carra and colleagues [48] provided direct experimental evidence supporting the role of ventilatory work in the development of SC_amp_. A previous study reported that respiratory muscle training resulted in a disappearance of SC_amp_ in obese patients during heavy-intensity exercise [36]. When DL-HVY was performed at POST, the same power output provoked a lower $\dot{V}$_E_ and a reduced $\dot{V}$O_2-RM_, i.e., a reduced O_2_cost of breathing. Despite this, the reduced $\dot{V}$O_2-RM_, *per se*, did not explain the reduction of SC_amp_. Indeed, the slopes of the corrected $\dot{V}$O_2p_ vs. time in SL-ET_GRP_ and DL-ET_GRP_ were not significantly different from the ones assessed from the “gross” $\dot{V}$O_2p_, thus confirming the muscular contribution to the modulation of $\dot{V}$O_2pSC_ in our volunteers. However, the lower $\dot{V}$_E_ and $\dot{V}$O_2-RM_, together with the reduced $\dot{V}$O_2p_, HR at the end of exercise, and [La]^b^_peak_ confirm an enhanced tolerance to intense exercise.

Also, $\dot{Q}$_m_O_2_ adaptations can influence $\dot{V}$O_2p_ response during heavy intensity exercise. In our study, we were unable to investigate if a faster increase of $\dot{Q}$_m_O_2_ induced the beneficial adaptations leading to the reduction of SC_amp_. To our knowledge, no studies have investigated in detail the mechanisms underpinning the modulation of SC_amp_ induced by ET in HTx, KTx, and LTx, making unfortunately impossible a comparison with other studies dealing with Tx.

In the present study, the lack of muscle biopsies to assess the possible increase in capillary and mitochondrial densities, endothelial function, enzyme activities, and capillary density prevented us from discerning the effect of $\dot{Q}$_m_O_2_ vs. m$\dot{V}$O_2_ on $\dot{V}$O_2p_ kinetics. Another limitation is that $\dot{Q}$_a_O_2_ kinetics was not assessed. Yet, HR kinetics was suggested to be a good proxy for the adjustment of O_2_ delivery to the imposed work rate [42]. To identify the power output associated with the $\dot{V}$O_2_-VT1, the $\dot{V}$O_2p_ vs. time relationship was left-shifted by 30 sec, instead of using an amount of time corresponding to the individual mean response time of the $\dot{V}$O_2p_ kinetics. This approach may have caused the overestimation of the work rate used for the DL-MOD and DL-HVY. However, our data confirm that our patients exercised in the moderate and heavy intensity domains. Finally, the absence of an age-matched control group precluded the determination of potential differences, compared with healthy mates, in the adaptive mechanisms elicited by the two diverse exercise training modalities.

## 5. Conclusions

In conclusion, eight weeks of DL-ET and SL-ET significantly speeded $\dot{V}$O_2p_ and HR kinetics and reduced O_2_Def during moderate intensity exercise carried out recruiting large muscle masses. Besides, the two training modalities resulted in the attenuation of the SC_amp_, suggesting that SL-ET is as effective as DL-ET when we aim to improve exercise capacity during heavy–intensity exercise. More studies are required to evaluate the relative contributions of the amelioration of $\dot{Q}$_m_O_2_ and of the peripheral gas exchanges in inducing the observed improvement of $\dot{V}$O_2p_ kinetics in Tx when the effects of small muscle mass training are compared to those of more traditional endurance training modalities.

## Figures and Tables

**Figure 1 ijerph-19-09097-f001:**
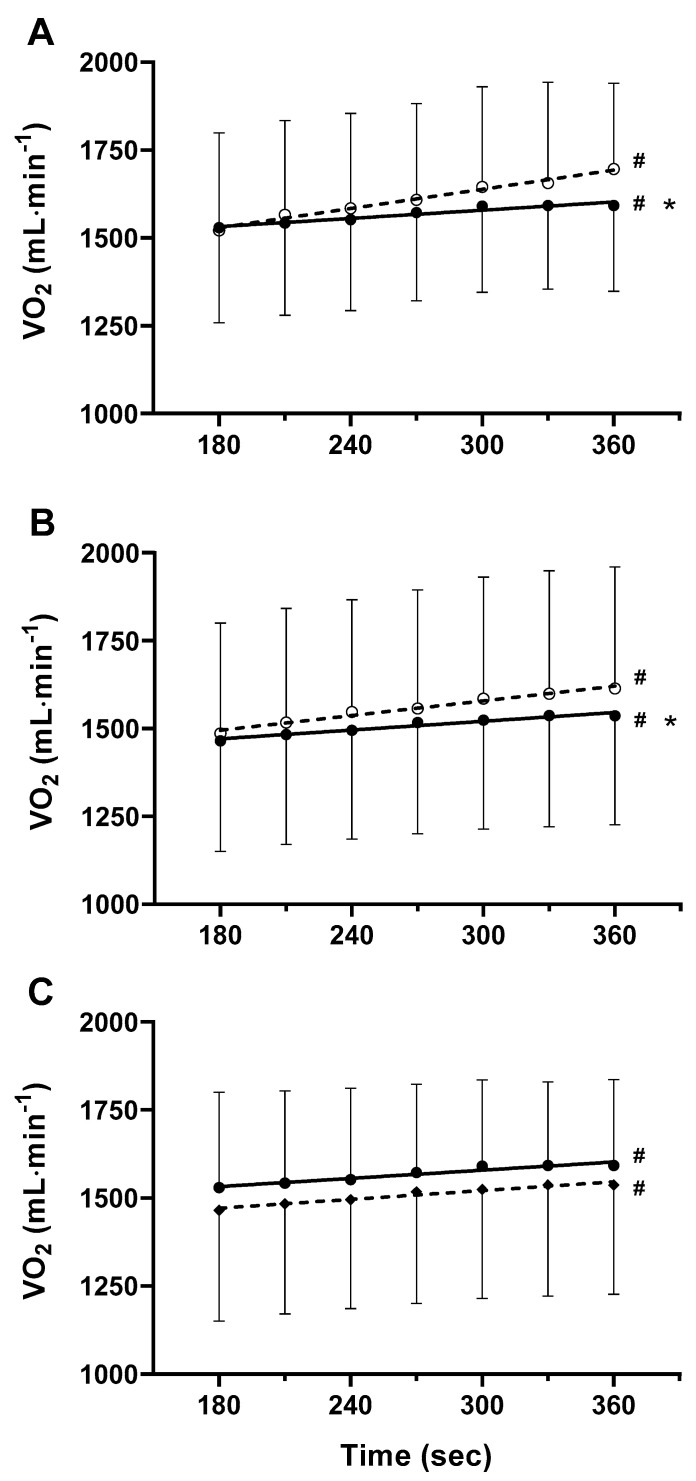
$\dot{V}$O_2p_ vs. time from the 3rd to 6th min of exercise during DL-HVY. Data show the linear increase of $\dot{V}$O_2p_ vs. time before (PRE, empty circles) and after training (POST, black points) in SL-ET_GRP_ (**A**) and in DL-ET_GRP_ (**B**). Data in (**C**) refers to the linear increase of $\dot{V}$O_2p_ vs. time for SL-ET_GRP_ (black circles) and for DL-ET_GRP_ (black diamond) at POST. Data are expressed as mean ± standard deviation. #: angular coefficient of the regression line is significantly different from zero. *: angular coefficient of the regression line is significantly different between PRE and POST.

**Figure 2 ijerph-19-09097-f002:**
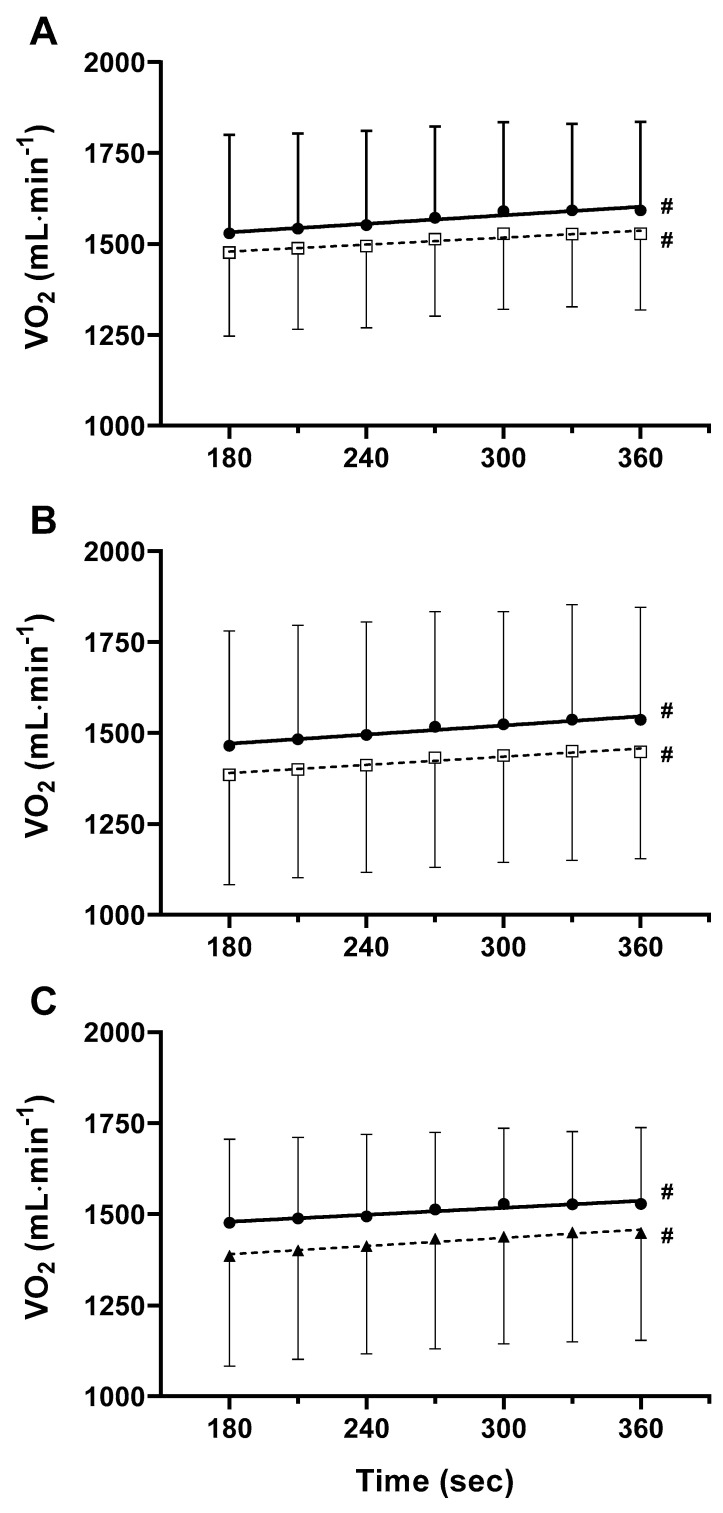
$\dot{V}$O_2_ vs. time from the 3rd to the 6th min of exercise during DL-HVY. Data in (**A**) show the relationship between the $\dot{V}$O_2p_ diminished by the estimated O_2_ cost of breathing (white squares) and the gross $\dot{V}$O_2_ (black circle) vs. time SL-ET_GRP_ after training. Data in (**B**) show the same relationship as in (**A**), but in DL-ET_GRP_. Data in (**C**) refers to the linear increase of $\dot{V}$O_2p_ subtracted by the estimated O_2_ cost of breathing for SL-ET_GRP_ (black circles) compared to DL-ET_GRP_ (black triangle). Data are expressed as mean ± standard deviation. #: angular coefficient of the regression line is significantly different from zero.

**Table 1 ijerph-19-09097-t001:** Main anthropometrics, cardio-respiratory, and cardiovascular parameters assessed during the double leg incremental test measured before the training period. Single leg endurance training group (SL-ET_GRP_) and double leg endurance training group (DL-ET_GRP_) groups.

*Anthropometrics*	SL-ET_GRP_ (*n* = 17)	DL-ET_GRP_ (*n* = 16)	*p*
Age (years)	56 (10)	55 (10)	0.82
BM (Kg)	83 (15)	78(18)	0.35
BMI	26.7 (3.3)	26.3 (5.3)	0.76
Years post-transplant	6.2 (6.9)	8.9 (7.7)	0.30
*Cardio-respiratory parameters*			
$\dot{V}$O_2peak_ (mL·min^−1^)	1747 (420)	1719 (483)	0.86
$\dot{V}$O_2peak_ (mL·kg^−1^·min^−1^)	21.8 (7.7)	22.2 (4.7)	0.85
$\dot{V}$_Epeak_ (L·min^−1^)	78.6 (16.9)	77.5 (21.3)	0.88
$\dot{V}$O_2_ at VT1 (mL·min^−1^)	1311 (216)	1275 (290)	0.55
HR_peak_ (bpm)	124 (24)	143 (22)	0.03
SpO_2_ (%)	96 (2)	96 (3)	0.93
Peak power output ($\dot{W}$)	133 (32)	132 (33)	0.95
Power output ($\dot{W}$) at VT1	76 (15)	76 (20)	0.87

Values are expressed as mean ± standard deviation. Abbreviations: BMI = body mass index; DL-ET_GRP_ = double leg endurance training group; HR = heart rate; SL-ET_GRP_ = single leg endurance training group; SpO_2_ = percent saturation of oxyhaemoglobin; $\dot{V}$_E_ = minute ventilation; $\dot{V}$O_2p_ = pulmonary O_2_ uptake; VT1 = first ventilatory threshold.

**Table 2 ijerph-19-09097-t002:** Pharmacological therapies.

Medications	SL-ET_GRP_ (n: 17)	DL-ET_GRP_ (n: 16)
Immunosuppressant	17 (100%)	16 (100%)
Corticosteroids	7 (41%)	4 (25%)
NSAID	10 (59%)	9 (56%)
ACE-inhibitors	2 (12%)	1 (6%)
Angiotensin 2 receptor blockers	4 (23%)	1 (6%)
α-blockers	6 (35%)	4 (25%)
β-blockers	11 (65%)	4 (25%)
Diuretics	3 (18%)	2 (13%)
Calcium channel blockers	4 (23%)	3 (19%)
Statins	6 (35%)	3 (19%)
Lipid lowering agents	2 (12%)	0 (0%)
Metformin	1 (6%)	1 (6%)
Insulin	1 (6%)	0 (0%)
Thyroid hormones	1 (6%)	4 (25%)
Proton pump inhibitors	10 (59%)	6 (38%)
Xanthine oxidase inhibitors	3 (18%)	4 (25%)
Hypouricemic agents	7 (41%)	5 (31%)
Kinase inhibitor agents	1 (6%)	2 (13%)
Bisphosphonates	1 (6%)	1 (6%)
Dopamine agonists	1 (6%)	0 (0%)
Bronchodilators	1 (6%)	0 (0%)
Antigout agents	1 (6%)	1 (6%)
Antiarrhythmic agents	1 (6%)	0 (0%)

Non-steroidal anti-inflammatory drugs (NSAID), dingle leg endurance training group (SL-ET_GRP_), Double leg endurance training group (DL-ET_GRP_).

**Table 3 ijerph-19-09097-t003:** Pulmonary O_2_ uptake ($\dot{V}$O_2p_) and heart rate (HR) kinetics parameters assessed during double leg moderate constant load exercise (DL-MOD) before (PRE) and after (POST) endurance training period. Single leg endurance training group (SL-ET_GRP_) and double leg endurance training group (DL-ET_GRP_) groups.

	SL-ET_GRP_ (*n* = 14)			DL-ET_GRP_ (*n* = 13)			Effect Size	*** *p* Values		
$\dot{V}$O_2p_ Kinetics	PRE	POST	Mean Difference (95% CI)	PRE	POST	Mean Difference (95% CI)	ηp^2^	G	T	G × T
O_2_ Def (mL O_2_)	728 (168)	596 (131) ^†^	132 (31; 233)	734 (278)	537 (204) ^†^	197 (101; 293)	0.54	0.707	<0.0001	0.276
MRT (s)	52.1 (15.9)	43.5 (15.2) ^†^	8.5 (3 to 14)	53.3 (14.4)	38.6 (9.5) ^†^	14.6 (9; 20)	0.63	0.707	<0.0001	0.083
SC_amp_ (mL O_2_)	207 (57)	131 (68) ^†^	75.5 (27; 124)	207 (84)	118 (90) ^†^	90 (40; 138)	0.51	0.767	<0.0001	0.645
HR kinetics										
Baseline	71.6 (11.6) ^▲^	70.3 (10.3)	1.3 (−4.2; 6.8)	81.2 (10.7)	78.3 (12.3)	3 (−2.3; 8.3)	0.07	0.040	0.193	0.615
Amplitude	21.6 (6.9) ^▲^	19.6 (6.6)	2 (−1; 5)	26.4 (6)	22.9 (5.5) ^†^	3.5 (0.7; 6.4)	0.29	0.082	0.038	0.375
Time delay (s)	13.4 (10.7)	17.8 (19.5)	−4.4 (−12.3; 3.5)	11.9 (13.7)	12.3 (13.8)	−0.4 (−8.2; 7.3)	0.04	0.510	0.312	0.403
Time constant (s)	78.4 (79.9)	50.6 (59) ^†^	27.7 (11.8; 43.7)	62.9 (36.4)	41.9 (28.7) ^†^	21 (5.7; 36.2)	0.53	0.557	<0.0001	0.468
MRT (s)	91.7 (87)	68.4 (65.3) ^†^	23.3 (8.2; 38.4)	74.8 (48.1)	54.2 (39.2) ^†^	20.5 (6.2; 34.9)	0.51	0.513	<0.0001	0.655
95% CI for time constant	58–81	39–55	-	52–65	37–48		-	-	-	-

Values are expressed as mean ± standard deviation, note that $\dot{V}$O_2p_ slow component amplitude (SC_amp_) refers to double leg heavy constant load exercise. ***: *p*-values from the two-way ANOVA are listed as group effect (G), time effect (T), groups × time effect (G × T). ^†^: Post-hoc test identifies significance (*p* ≤ 0.05) in differences between PRE and POST. ^▲^: Post-hoc test identifies a significant trend (0.05 < *p* ≤ 0.1) between groups at PRE.

**Table 4 ijerph-19-09097-t004:** Main cardio-respiratory parameters assessed during double leg moderate constant load exercise (DL-MOD) before (PRE) and after (POST) endurance training period. Single leg endurance training group (SL-ET_GRP_) and double leg endurance training group (DL-ET_GRP_) groups.

DL-MOD	SL-ET_GRP_ (*n* = 14)			DL-ET_GRP_ (*n* = 15)			Effect Size	*** *p* Values		
Steady-State Parameters	PRE	POST	Mean Difference (95% CI)	PRE	POST	Mean Difference (95% CI)	ηp^2^	G	T	G × T
$\dot{V}$O_2p-ss_ (mL·min^−1^)	1206 (173)	1180 (169)	25 (−11; 61)	1135 (226)	1131 (229)	4 (−30; 40)	0.07	0.424	0.169	0.328
$\dot{V}$CO_2p-ss_ (mL·min^−1^)	1153 (154)	1090 (141) ^†^	62 (20 to 104)	1060 (197)	1019 (201) ^†^	41 (1 to 81)	0.40	0.215	0.0002	0.398
$\dot{V}$_E-ss_ (L·min^−1^)	41.8 (4.4)	39.8 (4.9) ^#^	2 (−0.1 to 4.1)	38.5 (8.7)	36.1 (6.9) ^†^	2.4 (0.4; 4.4)	0.32	0.154	0.001	0.734
RER	0.96 (0.04)	0.93 (0.04) ^†^	0.03 (0.01; 0.06)	0.94 (0.04)	0.90 (0.03) ^†^	0.03 (0.01; 0.06)	0.40	0.080	0.0002	0.873
Gross Efficiency (%)	14.7 (1.2)	15 (1.4)	−0.3 (−0.8 to 0.1)	14.4 (2.3)	14.5 (2.1)	0.1 (−0.5 to 0.3)	0.11	0.595	0.100	0.404
O_2_ cost (mL·Watt^−1^)	14.0 (1.1)	13.9 (1)	0.1 (−0. to 0.7)	14.4 (3.2)	14.4 (2.4)	0 (−0.6 to 0.6)	0.00	0.524	0.727	0.807
HR_ss_ (bpm)	97 (17) ^⁑^	91 (15) ^†^	5 (1 to 9)	111 (12)	102 (12) ^†^	9 (5; 13)	0.58	0.022	<0.0001	0.094
[La]^b^ (mmol·L^−1^)	3.15 (1.11)	2.45 (1.15) ^†^	0.7 (0.3 to 1.1)	3.18 (0.84)	1.98 (0.75) ^†^	1.15 (0.8; 1.5)	0.76	0.689	<0.0001	0.057
Power ($\dot{W}$)	62 (13)	62 (13)	-	58 (14)	58 (14)	-	-	-	-	-

Values are expressed as mean ± standard deviation. ***: *p*-values from the two-way ANOVA are listed as group effect (G), time effect (T), groups × time effect (G × T). ^#^: Post-hoc test identifies a significant trend (0.05 < *p* ≤ 0.1) in differences between PRE and POST. ^†^: Post-hoc test identifies significance (*p* ≤ 0.05) in differences between PRE and POST. ^⁑^: Post-hoc test identifies significance in differences between groups at PRE.

**Table 5 ijerph-19-09097-t005:** Main cardio-respiratory parameters assessed during double leg heavy constant load exercise (DL-HVY) before (PRE) and after (POST) endurance training period. Single leg endurance training group (SL-ET_GRP_) and double leg endurance training group (DL-ET_GRP_) groups.

DL-HVY	SL-ET_GRP_ (*n* = 17)			DL-ET_GRP_ (*n* = 16)			Effect Size	*p* Values ***		
End-Exercise Parameters	PRE	POST	Mean Difference (95% CI)	PRE	POST	Mean Difference (95% CI)	ηp^2^	G	T	G × T
$\dot{V}$O_2p_ (mL·min^−1^)	1712 (293)	1636 (249) ^†^	76 (15; 137)	1665 (344)	1563 (351) ^†^	103 (40; 166)	0.43	0.579	<0.0001	0.469
$\dot{V}$CO_2p_ (mL·min^−1^)	1651 (263)	1554 (238) ^†^	97 (37; 158)	1645 (334)	1478 (329) ^†^	167(104; 229)	0.62	0.689	<0.0001	0.069
RER	0.97 (0.04)	0.95 (0.03) ^#^	0.02 (0; 0.04)	0.99 (0.06)	0.95 (0.04) ^†^	0.04 (0.02; 0.07)	0.40	0.540	<0.0001	0.067
O_2_ cost (mL·Watt^−1^)	14.2 (1.2)	13.6 (1.4)	0.6 (−0.2; 1.4)	14.8 (2.9)	13.5 (1.7) ^†^	1.3 (0.5; 2.1)	0.33	0.635	<0.0005	0.158
$\dot{V}$_E_ (L·min^−1^)	70.3 (13.3)	62.8 (10.2) ^†^	7.4 (2.9; 12)	68.0 (13.8)	56.5 (13.3) ^†^	11.5 (6.8; 16.2)	0.60	0.313	<0.0001	0.158
$\dot{V}$O_2-RM_	116 (27)	104 (24) ^†^	12 (1.5; 22.7)	113 (29)	93 (24) ^†^	20 (9; 31)	0.44	0.394	<0.0001	0.230
HR (bpm)	126 (24)	118 (22) ^†^	8 (3; 13)	141 (18)	128 (19) ^†^	13 (8; 18)	0.60	0.098	<0.0001	0.137
[La]^b^ (mmol·L^−1^)	6.2	5.2 ^†^	1 (0.1; 2)	6.6	4.6 ^†^	2 (1.1; 2.9)	0.52	0.881	<0.0001	0.088
RPE_dyspena_	15.1 (1.5)	13.6 (1.6) ^†^	1.5 (0.1; 2.9)	16.2 (1.6)	14.1 (1.8) ^†^	2.1 (0.7; 3.4)	0.40	0.072	0.0001	0.523
RPE_leg pain_	6.2 (1.6)	5.1 (1.7) ^†^	1.1 (0.1; 2.2)	6.3 (1.4)	5.1 (1.9) ^†^	1.2 (0.1; 2.2)	0.32	0.964	0.001	0.932
Power ($\dot{W}$)	95 (24)	95 (24)	-	94 (25)	94 (25)	-	-	-	-	-

Values are expressed as mean ± standard deviation. ***: *p*-values from the two-way ANOVA are listed as group effect (G), time effect (T), groups × time effect (G × T). ^†^: Post-hoc test identifies significance (*p* ≤ 0.05) in differences between PRE and POST. ^#^: Post-hoc test identifies a significant trend (0.05 < *p* ≤ 0.1) in differences between PRE and POST.

## Data Availability

The data presented in this study are available on request from the corresponding author.

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
