# Peer review of "The Effect of Endurance Training on Pulmonary V˙O2 Kinetics in Solid Organs Transplanted Recipients"

_ijerph, 2022, doi:10.3390/ijerph19159097_

Round 1

Reviewer 1 Report

1.      Reviewed article submitted contain the required sections: Authors Information, Abstract, Keywords, Introduction, Materials & Methods, Results, Conclusions, Figures and Tables with Captions, Funding Information, Author Contributions, Conflict of Interest and other Ethics Statements.

2.      The results of the conducted research correspond to the set goal, the materials and research methods used are described in detail, statistical methods correspond to the purpose of the study.

3.      The article is written in a clear language, the data presented in the tables are correct, reflect the scientific results presented in the text.

4.      The obtained results of the studies presented in this article are undoubtedly characterized by high relevance, as they provide advancement and obtaining new data on the issues of rehabilitation of patients after parenchymatous organ transplantation.

5.      Numerous long-term studies have shown that recipients of parenchymatous organ transplantation experience decreased physical activity for many years after transplantation. Low physical activity tolerance is known to affect patients' quality of life, return to work, and active social life, and is also associated with increased transplant mortality.

6.      The results of this study expand knowledge regarding the implementation of rehabilitation measures in parenchymal organ transplant recipients.

7.      Increased knowledge regarding physical activity and its health benefits may be a step toward changing and supporting positive behavior in post-transplant patients.

8.      The citations cited need to be updated because only 6 of them are recent (within the last 5 years).

9.      The findings are logically consistent with the results presented.

10.  The data presented may be of significant interest to both a professional audience with regard to rehabilitation interventions in patients after parenchymal organ transplantation, and to a wide range of readers of the journal.

Reviewer 2 Report

Thank you for this article to review. I believe there is considerable merit to this work, however, I have a few suggestions and questions.

The intro is long and rambling, and at times it is unclear what the main point(s) are. Writing needs a lot of work.

There is Frequent use of hyperbole or unnecessary adjectives.

Materials and methods are written much better.

Line 121: two exercise sessions separated by at least 15-30 min. Would it be more appropriate to say ‘by at least 15 min’ as that appears to be the shortest possible range, and assuming some were more than 30 min. Or, were all sessions separated by 15-30 min, no more or less?

Sec 2.3 states that WU was done at 25W or 40W, and incremental exercise went up 15W per minute to exhaustion. The proportional increase between the two warmup groups is excessively different, with the 25w group increasing initially 60% while the 40w group only increased initially ~37%. Why didn’t you have proportionately similar incremental increases?

Line 131 discusses the cooldown, which was done at 25w. Again, this seems odd, as for one group that was equivalent to their WU, but for the other group it was at a much lower workrate?

Pedaling has only on ‘l’.

Line 186 states that during ET, pedaling cadence was held between 60-70 rpm, yet in each of the test protocols it is listed as 60-75 rpm, why the difference?

In Table 1, it appears there is a significant difference between the peak HR of the two groups (p=0.03), yet there is nothing to indicate that a difference exists?

Table 3 doesn't seem necessary??

Line 350 states that all DL participants ‘terminated the test at POST, whereas only one SL terminated at the 7th minute’. Can you be more clear on this. Are you trying to say that some volunteered to stop, while others made it to a predetermined endpoint and then stopped?

Line 403: very confusing statement, plus the design was intentionally heterogeneous, so I am not sure why you state 'somehow' heterogeneous?

Line 442 is difficult to understand.

Reviewer 3 Report

Thank you for the opportunity to revise the current manuscript. The authors have done good work and I only have some minor comments regrading the introduction section. Please see my comments below,

Line 4 Mates of similar age ? Please explain this

Line 31 Is this the first time you are using this abbreviation ? If so, please define ?̇mO2

Line 51 This is quite interesting, could you please explain or propose a physiological mechanism governing these adjustments?
